# AMF Inoculum Enhances Crop Yields of *Zea mays* L. ‘Chenghai No. 618’ and *Glycine max* L. ‘Zhonghuang No. 17’ without Disturbing Native Fugal Communities in Coal Mine Dump

**DOI:** 10.3390/ijerph192417058

**Published:** 2022-12-19

**Authors:** Kun Wang, Yinli Bi, Jiayu Zhang, Shaopeng Ma

**Affiliations:** 1State Key Laboratory of Coal Resources and Safe Mining, China University of Mining and Technology (Beijing), Beijing 100083, China; 2Institute of Ecological Environment Restoration in Mine Areas of West China, Xi’an University of Science and Technology, Xi’an 710054, China

**Keywords:** AMF community composition, intercropping, coal mining dump, bioinvasive risk, crop yields

## Abstract

For the agricultural development of dumps, increase in land use efficiency and protection of food security, to verify the safety, efficacy and sustainability of field-applied arbuscular mycorrhizal fungi (AMF) inoculum, and to exclude the risk of potential biological invasion, in this study, we determined the effect of AMF inoculation and intercropping patterns (maize–soybean) on the temporal dynamics of soil parameters, native AMF communities and crop yields. AMF communities were analyzed using Illumina MiSeq. A total of 448 AMF operational taxonomic units (OTUs) belonging to six genera and nine families were identified. AMF inoculation treatment significantly improved the yield of intercropping maize and increased the content of available phosphorus. AMF diversity was significantly influenced by cropping pattern and growth stage, but not by the inoculation treatment. Inoculation altered the AMF community composition in the early growth stage and facilitated a more complex AMF network in the early and late growth stages. These results indicate that AMF inoculation affects native AMF only in the early stage, and its impact on yield may be the consequence of cumulative effects due to the advantages of plant growth and nutrient uptake in the early stage.

## 1. Introduction

The Anthropocene, the present geological epoch defined by human footprints, has wreaked havoc on the ecosystem and jeopardized the global food supply [1]. Coal mining results in an abundance of ecological and environmental problems. Due to excessive coal mining, vast areas of land and flora have been destroyed, resulting in changes to the soil structure and physicochemical qualities of the soil, as well as the construction of large dump sites [2,3]. However, during the process of dump formation, the original topography was reshaped to form a flat platform, e.g., a “terraced field”, which is suitable for agricultural development and increases in grain production. In our previous studies, we showed that soil nutrients and microbial communities were restored to a greater extent in dumps with more than 10 years of reclamation [4,5]. Therefore, these areas may meet the conditions for agricultural development. However, extensive use of chemical fertilizers and water in traditional agriculture is likely to cause secondary damage to the reclaimed land. Thus, it is important to use more efficient fertilization programs and agricultural management models for agricultural development in these areas.

Given the economic and environmental expenses associated with irrigation and chemical fertilization, contemporary agriculture should use microbiomes to optimize production while minimizing input in the case of future environmental upheavals [6,7]. Traditional large-scale agricultural operations place a heavy emphasis on water and chemical fertilizer, while microbiomes play an important role in boosting plant absorption of soil water and nutrients [6,8]. As well as contributing to increased crop yields, Arbuscular mycorrhizal fungi (AMF) play a significant role in ecosystems (e.g., soil structure, nutrient conservation, and plant stability in changing environments) and may lower the quantity of fertilizer necessary for cost effectiveness [9].

AMF are a significant group of root-associated mutualists in the plant microbiome, and they may develop mutualistic partnerships with over 80% of terrestrial plant species [10]. AMF exchange soil-derived nutrients for photosynthates from the host plant in this relationship [10]. AMF relationships have been found in both field and laboratory studies to improve soil nutrient status and plant growth in post-mining environments [11,12]. Because mycorrhizal associations can range from mutualism to parasitism depending on environmental and species-specific factors [13], AMF could help to encourage more mutualistic or parasitic partnerships. Currently, there is considerable controversy among researchers about how native AMF communities respond to the addition of exogenous AMF inoculum [14,15] and whether the application of AMF inoculum poses a bioinvasive risk to the regional ecology [16,17]. Some studies have shown that AMF inoculation has little or no effect on native fungal communities [18,19], whereas others have shown that exogenous AMF inoculum can displace dominant native microbial taxa and cause disturbance to native microbial communities [20,21]. Scholars are more concerned about the negative impact on plant productivity due to the reduction of soil biodiversity caused by the spread of inoculated AMF to non-target areas [22,23]. Therefore, it is necessary to assess the changes in native AMF communities after the application of AMF inoculum.

Intercropping, often known as polyculture or mixed cropping, is a common farming strategy used across the world to prevent soil-borne plant pathogens from accumulating [24]. Intercropping strategies can improve the ecosystem’s agro-quality while also assisting in the management of diseases, weeds, and pests [25,26]. This is mostly accomplished by antagonistic secondary metabolites produced by one plant root that successfully inhibits the pathogen of another plant [27]. Increased yield, production sustainability, ecosystem development, and environmental safety are all benefits of the intercropping system. In an intercropping system, two or more crop species are cultivated at the same time; they cohabit and interact with one another and with the agro-ecosystems [26].

Microbes’ temporal dynamics have been exploited to identify elements that influence community organization and ecological functions [28,29,30]. Shifts in the diversity and community structure of AMF assemblages across time and space are linked to plant community succession, anthropogenic activities, and changes in environmental conditions [31,32,33,34,35].

Current understanding of the effects of AMF inoculation is mainly based on short-term greenhouse experiments, with more research on plant performance before and after AMF inoculation [16], and less research on the sustainability of AMF inoculants under field application conditions. To determine the safety, sustainability and effectiveness of using AMF as biofertilizer in the agricultural development of open-pit coal mine dumps and to exclude potential environmental risks and hazards, this study investigated the effects of AMF inoculum on the community of native AMF and its coupling with soil factors under different cropping patterns (monocrop and intercrop) during three periods of crop growth. This study aims to improve crop yields and soil fertility by using biofertilizers that are beneficial to the environment. The economic, ecological, and social benefits of coal mine dumps can be achieved through the production of green, organic, and healthy products for human consumption.

## 2. Materials and Methods

### 2.1. Study Area

The soil sampled for this research came from an opencast coal mine dump in the Heidaigou mining region, Jungar Banner, Ordos City, Inner Mongolia, northern China (N 39°43′–39°49′, E 111°13′–111°20′). The soil texture is loess and covers the whole coal field. The region is characterized by a typical medium-temperate semi-arid continental climate, with an annual mean temperature of 7.2 °C, a maximum temperature of 38.3 °C, and a minimum temperature of −30.9 °C. The four seasons have unique climatic conditions. Annual precipitation totals 231−460 mm, with a mean of 404 mm. July through September receives most of the precipitation, accounting for around 60% to 70% of the total annual precipitation. The yearly mean evaporation is 2082 mm, and the annual mean sunlight hours are 3119.3 h. The range of soil moisture during the crop growth period is 7.2–23.8%, with a mean value of 13.8%.

### 2.2. Experimental Design

The experiment was initiated in the growing season of 2018–2019 at Heidaigou mining area north dump 1260 platform west. The research used a two-factor split-plot design with three replicates. The first factor was the inoculation treatment, which was designated as CK treatment (control treatment without AMF inoculants) and AMF treatment (with AMF inoculants). The AMF inoculants were created by inoculating sandy soils with *Funneliformis mosseae* (a type of AMF) spores and extraradical mycelium generated in pot cultures by maize plants. Extraradical mycelium had a density of 4.66 m·g^−1^ soil and spores had a density of 66 spores·g^−1^ soil. The second factor was the cropping patterns, which included: (i) monocropping of maize, (ii) monocropping of soybeans, and (iii) intercropping of maize and soybeans. The plots were randomly arranged under the same inoculation treatment. In total, there were 6 experimental treatments, including 2 inoculation treatments and 3 cropping patterns, resulting in a total of 24 plots (four replicates for each plot) in this study. 

Each plot had an area of 10 m (width) × 20 m (length) (some plots had increased in size due to the topography, Figure 1). Two rows of maize intercropped with four rows of soybeans formed one strip in the intercropping plots, and each intercropping plot had two strips. *Zea mays* L. ‘Chenghai No. 618’, and *Glycine max* L. ‘Zhonghuang No. 17’ were selected for the study. Before sowing, a total of 80 kg ha^−1^ K_2_O (as K_2_SO_4_) and 120 kg ha^−1^ P_2_O_5_ [as Ca(H_2_PO_4_)_2_] was applied to each plot. 

### 2.3. Soil Sampling

In 2019, soil samples were collected on 15 June (jointing stage), 20 July (flowering stage), and 3 September (harvesting stage). Five soil cores (3.5 cm in diameter and 20 cm in depth) were randomly taken from the area immediately next to the plant roots in each plot and were mixed to create a composite sample for analysis. Soil samples for DNA analysis were transported from the field to the laboratory on ice, and then stored at −80 °C before being processed for DNA extraction and high-throughput sequencing. All the other samples were sieved (<2 mm), air dried and stored at 4 °C before processing for the determination of physicochemical soil properties. A total of 96 samples (two inoculation treatments × four cropping patterns × three sampling growth stages × four replicates) were collected.

### 2.4. Edaphic Variables

Soil electrical conductivity (EC) and pH were determined using a glass electrode in the [1: 5 (*w*/*w*)] suspension with a conductivity meter (DDS-307W; Shanghai Lida Instrument Factory, Shanghai, China) or in the soil–water suspension [1: 2.5 (*w*/*w*)] with a pH meter (PHS-3C; Shanghai Lida Instrument Factory, Shanghai, China). Alkaline phosphatase (ALP) was measured using the methods described by Tarafdar and Marschner [36]. Total organic carbon (TOC) was determined using a Vario Max element analyzer (Vario Max, Elementar, Langenselbold, Hesse, Germany). Available phosphorus (P) and available potassium (K) were assessed by inductively coupled plasma–optical emission spectrometry (ICP-OES, Optima 5300DV; Waltham, MA, USA). NO_3_^−^-N and NH_4_^+^-N were determined using a Continuous Segmented Flow Analyzer (SEAL, AutoAnalyzer 3; Norderstedt, Germany). Glomalin was extracted from the soil samples, and the levels of easily extractable glomalin (EEG) were assessed using a modified version of the Wright and Janos procedure [37,38].

### 2.5. Molecular Identification of AMF

A MoBio PowerSoil DNA Isolation Kit (Mo Bio Laboratories, Carlsbad, CA, USA) was used to extract soil microbial DNA from 0.1 g homogenized soil from each sample according to the manufacturer’s instructions. The universal eukaryotic primer GeoA2 (5′-CCA GTA GTC ATA TGC TTG TCTC-3′) [39] and the specific primer AML2 (5′-GAA CCC AAA CAC TTT GGT TTC C-3′) [40] were used for the first round of PCR by targeting a small subunit (SSU) rDNA gene (yielding approximately 1000-bp amplicons), whereas primer pairs NS31 (5′-TTG GAG GGC AAG TCT GGT GCC-3′) [41] and AMDGR (5′-CCC AAC TAT CCC TAT TAA TCA T-3′) [42] with barcodes were used for the second round of PCR (approx. 300-bp amplicons). After mixing the PCR products in similar density ratios, they were purified. Following that, the library was constructed and the quality and quantity of the samples were determined using a fluorometer (Qubit 2.0, Thermo Fisher Scientific, Waltham, MA, USA) and a bioanalyzer system (2100, Agilent Corporation, Santa Clara, CA, USA). The amplicons were purified and sequenced on an Illumina MiSeq PE300 platform (Illumina Corporation, San Diego, CA, USA).

### 2.6. Bioinformatics Analysis

The quality of the raw sequences was analyzed by FASTQC, and then sequences were calculated using QIIME 2 v2019.7 [43]. Firstly, primer sequences were removed from the raw sequences using the Cutadapt v1.9.1 [44], and low-quality reads were filtered out by excluding ambiguous bases with an average quality score < 20 or read length < 200 bp. Secondly, sequences were denoised into operational taxonomic units (OTUs) by DADA2 [45]. Finally, the representative sequence for each OTU was screened for further annotation using a custom taxonomic classifier to gather taxonomic information. The sequence was chosen and compared to MaarjAM’s 18S rRNA gene database [46].

### 2.7. Statistical Analysis

We estimated and visualized diversity indices, including the number of observed species (Sobs), the Shannon–Wiener index, the Simpson index, and phylogenetic diversity using R software (v.3.5.2, https://www.r-project.org/ (accessed on 10 May 2021)) via the “phyloseq” package [47]. Histograms created by R software (v.3.5.2) through the “ggalluvial” package were used to visualize OTUs [48]. The permutational multivariate analysis of variance (PERMANOVA) and non-metric multidimensional scaling (NMDS) analyses were conducted using R software (v.3.5.2) via the “vegan” package [49], and the “adonis” and “metaMDS” functions from the “vegan” package. The effect of edaphic variables on the composition of AMF communities was quantified using the “envfit” function from the “vegan” package. Principal coordinate analysis (PCA) based on Bray–Curtis dissimilarities and random forest analysis were estimated and visualized by R software via the “microeco” package [50]. Non-random co-occurrence analysis based on the SParse Inverse Covariance Estimation for Ecological Association Inference (SPIEC-EASI) method was estimated using R software via the “SpiecEasi” package [51]. Network analysis was carried out and feature values were estimated using Gephi software (v.0.9.2, https://gephi.org/ (accessed on 10 June 2021)). Spearman’s correlation coefficients were used to determine relationships between edaphic variables and AMF diversity indices. The relationship between AMF community composition, growth stages, inoculation treatments, cropping patterns, and both edaphic variables and fungal diversity index was analyzed by the Mantel test. Results from Spearman’s correlation analysis and the Mantel test were visualized using R software via the “ggcor” package. Significant differences between edaphic variables and fungal diversity indices were assessed by three-way analysis of variance (ANOVA) using R software. If the results from ANOVA were significant, comparisons of the means were examined by multiple pairwise comparisons using Tukey’s honestly significant difference tests (*p* < 0.05) on SPSS (v.20.0; IBM, Armonk, NY, USA).

## 3. Results

### 3.1. Effect of AMF Treatment and Cropping Pattern on the Yield of Maize and Soybean

The average yield of maize in all plots was 7.42 ± 1.88 (t ha^−1^), ranging from 5.60 ± 0.05 to 9.80 ± 0.04. The average yield of soybean in all plots was 0.68 ± 0.18 (t ha^−1^), ranging from 0.49 ± 0.07 to 0.85± 0.05 (Figure 2, mean ± SD, *n* = 3). Two-way ANOVA showed that cropping pattern, inoculation treatment and their interactions significantly influenced maize yield (*p* < 0.05); whereas, only cropping pattern significantly influenced soybean yield (*p* < 0.05). In the same cropping pattern, the yield of corn and soybean under AMF treatment was higher than that under CK treatment (*p* < 0.05). The yield of intercropping maize under AMF treatment was significantly higher than that under CK treatment (*p* < 0.05). Under the same inoculation treatment (with or without AMF treatment), the yield of maize in the intercropping group was significantly higher than that in the monocropping group, while the soybean yield in the monocropping group was significantly higher than that in the intercropping group (*p* < 0.05). These results indicate that AMF treatment and cropping pattern affect the yield of maize and soybean significantly.

### 3.2. Effect of AMF Treatment and Cropping Pattern on Edaphic Variables

Three-way ANOVA showed that the EC and NH_4_^+^-N were significantly affected by growth stage, cropping pattern, the interaction between growth stage and cropping patterns, the interaction between growth stage and inoculation treatment, the interaction between cropping patterns and inoculation treatment, and the interaction between growth stage, inoculation treatment and cropping patterns (Table 1; *p* < 0.05). AP and NO_3_^−^-N were significantly affected by growth stage, inoculation treatment, cropping pattern, and their interactions (*p* < 0.05). AP under the AM treatment was significantly higher than under the CK treatment in the first stage with some cropping patterns (Appendix A; *p* < 0.05). 

### 3.3. Effect of AMF Treatment and Cropping Pattern on the Native AMF

A total of 4,581,166 AMF sequences were obtained and matched to 448 AMF OTUs (MaarjAM database). With bracketed relative abundances, these sequences represented the following four families: Glomeraceae (83.4%), Claroideoglomeraceae (9.8%), Diversisporaceae (3.6%) and Paraglomeraceae (3.2%). These OTUs represented all four genera of the Glomeromycota division. Among the 448 OTUs, 3 belonged to *Rhizoglomus*, 47 belonged to *Claroideoglomus*, 62 belonged to *Diversispora*, 58 belonged to *Paraglomus*, 70 belonged to *Septoglomus* and 208 belonged to the *Glomus* genus. The frequency distribution histogram of AMF species is shown in Figure 3. 

The relative abundance of *Septoglomus viscosum* VTX00063 was decreased in the growth stages of monocropping maize, and intercropping soybean under AMF treatment and in monocropping maize, intercropping soybean and intercropping maize under CK treatment. The relative abundance of *Funneliformis mosseae* VTX00067 (inoculated AMF) was increased in the growth stages of monocropping maize and intercropping maize under CK treatment. The relative abundance of *Diversispora* sp. VTX00060 was increased in the growth stages of the monocropping maize, intercropping soybean and intercropping maize under AM treatment and in intercropping soybean maize under CK treatment, but it was decreased in the growth stages of monocropping soybean maize under CK treatment. The relative abundance of *Paraglomus* sp. VTX00446 was increased in the growth stages of the intercropping soybean and intercropping maize under CK treatment. The relative abundance of *Claroideoglomus* sp. VTX00193 was decreased in three growth stages of intercropping maize under AM treatment and in monocropping soybean and intercropping soybean under CK treatment, but it was increased in the growth stages of monocropping maize under CK treatment.

Random forest analysis was performed to identify the high-dimensional biomarkers of AMF communities at the species level at different growth stages, cropping patterns and under different inoculation treatments (Figure 4). The results showed that the relative abundance of the *Septoglomus viscosum* VTX00063 was higher in the first stage, *Glomus* sp. VTX00156, *Glomus* sp. VTX00409 and *Glomus* sp. VTX00195 were more abundant in the second stage, whereas *Glomus* sp. VTX00060, *Diversispora* sp. VTX00306 and *Glomus* sp. VTX00107 were more abundant in the final stage (Figure 4A). The relative abundance of the *Diversispora* sp. VTX00060 and *Glomus* sp. VTX00156 were higher in monocropping maize, *Diversispora* sp. VTX00306 was more abundant in intercropping maize, *Septoglomus viscosum* VTX00063 was more abundant in monocropping soybean, whereas *Paraglomus* sp. VTX00446 was more abundant in intercropping soybean (Figure 4B). The relative abundance of *Septoglomus viscosum* VTX00063, *Diversispora* sp. VTX00060 and *Glomus* sp. VTX00195 under AMF treatment was significantly higher than under CK treatment, but the reverse results were observed for *Glomus* sp. VTX00114, *Paraglomus* sp. VTX00446 and *Glomus* sp. VTX00100 (Figure 4C).

### 3.4. Effect of AMF Treatment and Cropping Pattern on the Diversity of AMF Communities

The average Chao1 index was 58.05 ± 14.00, ranging from 46.25 ± 4.11 to 77.75 ± 9.07 (Figure 5). The average Shannon–Wiener index was 2.46 ± 0.43, ranging from 1.69 ± 0.35 to 3.16 ± 0.10. The average Simpson index was 0.82 ± 0.08, ranging from 0.64 ± 0.10 to 0.92 ± 0.01. The average phylogenetic diversity was 3.30 ± 0.45, ranging from 2.59 ± 0.33 to 4.04 ± 0.14. Three-way ANOVA showed that the AMF Chao1 index and Shannon–Wiener index were significantly influenced by cropping pattern and growth stage (*p* < 0.05). The Simpson index was significantly influenced by cropping pattern, growth stage and the interaction between cropping patterns and growth stage (*p* < 0.05). Phylogenetic diversity was significantly affected by the interaction between growth stage and cropping patterns (*p* < 0.05, Figure 5). The AMF Chao1 index, Shannon–Wiener index, Simpson index and phylogenetic diversity were not significantly different at different growth stages under both cropping patterns. The AMF Chao1 index in intercropping maize was significantly higher than that in the monocropping soybean in the second stage under CK treatment. The AMF Chao1 index in the second stage was significantly higher than that in the first and final stages in monocropping and intercropping maize under both CK and AMF treatments. The AMF Shannon–Wiener index was significantly higher in the second stage than in the final stage in intercropping maize and soybean under both CK and AMF treatment. The AMF Shannon–Wiener index in the second stage was significantly higher than in the first and final stages in monocropping maize under AMF treatment. The Simpson index in the second stage was significantly higher than that in the final stage in intercropping soybean under AMF treatment. The temporal analysis showed that the AMF Chao1 index, the Shannon–Wiener index and the Simpson index increased in the second stage and decreased in the final stage under different cropping patterns and inoculation treatments (Figure 5). However, phylogenetic diversity did not differ significantly between different growth stages. These results indicate that the temporal dynamics of the AMF Chao1, Shannon–Wiener and Simpson indices were not influenced by inoculation treatments or cropping patterns.

PERMANOVA analysis showed that the AMF community composition was significantly affected by the growth stage (*pseudo*-F = 10.271, *R*^2^ = 0.092, *p* < 0.001), inoculation treatment (*pseudo*-F = 2.035, *R*^2^ = 0.018, *p* < 0.05), cropping patterns (*pseudo*-F = 3.989, *R*^2^ = 0.036, *p* < 0.01), the interaction between growth stage and inoculation treatment (*pseudo*-F = 2.113, *R*^2^ = 0.019, *p* < 0.05), and the interaction between growth stage and cropping patterns (*pseudo*-F = 2.878, *R*^2^ = 0.026, *p* < 0.01). Similarly, PCoA revealed that the composition of the AMF community was significantly different among different growth stages, inoculation treatments and cropping patterns (Figure 6). Furthermore, the AMF community composition in the first stage was significantly affected by inoculation treatment. AMF community composition was similar in intercropping maize and monocropping maize under CK treatment. It was also similar in intercropping maize and intercropping soybean under AMF treatment. By contrast, AMF community composition in the second stage was not significantly affected by inoculation treatment. AMF community composition differed significantly between the two cropping patterns under CK treatment. AMF community composition was similar in intercropping soybean and monocropping maize under AMF treatment. AMF community composition in final stage was not significantly affected by inoculation treatment.

### 3.5. Network Analyses

*Glomus*, *Septoglomus*, *Diversispora*, *Claroideoglomus* and *Paraglomus* were the main groups of the AMF network, accounting for 31.83–46.23%, 8.89–21.36%, 4.85–20.39%, 9.52–17.92% and 2.22–10.13%, respectively (Figure 7). In AMF treatment, the nodes percentage of *Diversispora* was increased with the growth stages, with the maximum nodes percentage observed in the first stage. The minimum value of AMF network nodes, edges, average degrees (avgK), average path distances (GD) and network diameter (90–103, 72–83, 1.59–1.61, 2.24–2.69, 6–10, respectively) in the AMF treatment occurred in the second stages. The maximum value of AMF network average clustering coefficient (avgCC) and heterogeneity (0.09–0.23, 0.38–0.47, respectively) in the AMF treatment occurred in the final stages and decreased with the growth stages. The AMF network modularity (0.79–0.91) in AM treatment had a maximum value in the final stage and increased with the growth stages. In CK treatment, the nodes percentage of *Claroideoglomus* was increased with the growth stages, with the maximum observed in the first stage. The maximum value of AMF network nodes, edges and heterogeneity (79–106, 59–87, 0.399–0.446, respectively) in CK treatment occurred in the second stage and the first stage, which was higher than those in the final stage. However, the AMF network avgK, GD, network diameter and centralization (1.45–1.64, 1.69–3.20, 4–9, 0.019–0.022, respectively) in CK treatment had minimum values at the second stage and the final stage, which was higher than those in the first stage.

### 3.6. Correlation Analyses

The envfit on the NMDS plot indicated that AMF community composition was significantly correlated with soil pH (*R*^2^ = 0.076), electrical conductivity (EC, *R*^2^ = 0.116), NH_4_^+^–N (*R*^2^ = 0.262), NO_3_^−^–N (*R*^2^ = 0.451), available phosphorus (AP, *R*^2^ = 0.181), available potassium (AK, *R*^2^ = 0.176), total organic carbon (TOC, *R*^2^ = 0.065), easily extractable glomalin (EEG, *R*^2^ = 0.216) and alkaline phosphatase (ALP, *R*^2^ = 0.111) (Figure 8a). The Mantel test and Spearman’s correlation test showed significant correlations among stages of edaphic variables, AMF diversity indices, AMF community composition, growth stages, inoculation treatment, and cropping patterns (Figure 8b). The results also showed that AMF community composition had significant effects on EC, pH and EEG (*p* < 0.05). Growth stage had significant effects on all edaphic variables (*p* < 0.05) and AMF diversity indices (*p* < 0.05) except for the phylogenetic diversity (PD). AMF treatment had significant effects on TOC, AP and AK (*p* < 0.05). Cropping patterns affected AK, NH_4_
^+^ –N and NO_3_^−^–N significantly (*p* < 0.05). AMF treatment and cropping pattern had no significant effects on the AMF diversity index (*p* ≥ 0.05). The AMF Chao1 index was positively correlated with soil EC and pH (*p* < 0.05), and negatively correlated with NH_4_^+^–N and NO_3_^−^–N (*p* < 0.05). Soil pH positively influenced the AM fungal Shannon–Wiener index and Simpson index (*p* < 0.05), whereas soil EEG was negatively correlated with the AMF Shannon–Wiener index and Simpson index (*p* < 0.05).

## 4. Discussion

In this study, we showed that AMF treatment increased the yield of maize and soybean, which is consistent with previous results showing that seed inoculation with selected isolates can increase plant root colonization and crop productivity [52,53,54]. The impact of AMF inoculation on soil microbial biodiversity has recently become a hot issue of discussion [15,55]. We showed that AMF treatment changed the native AMF community composition in the early stage of growth but had no significant effect in the middle and late stages of growth and did not affect AMF alpha diversity. These results indicate that the exogenous inoculants used in this study did not significantly affect the native AMF community. The interactions of various AMF isolates have been shown to be synergistic, neutral, or antagonistic [56,57,58]. Moreover, indigenous AMF, which is better adapted to the local conditions, can outcompete some of the inoculated fungi [59]. 

The dynamic link between plant dependency on mycorrhizal associations and nutrient availability, eutrophication, or growth-limiting circumstances is largely responsible for the influence of edaphic variables on AMF community structure [60]. Physical and chemical characteristics of the soil may also have a significant effect on the symbiotic connection between plants and fungus [61]. We found that the AMF inoculation significantly affected the soil total organic carbon, available P and available K, especially the content of available P. The establishment of mycorrhizal symbioses is helpful for the mobilization and absorption of phosphorus by plants in AP-limited soils [10]. In general, the synergy and complementarity of diverse modes of action might give additional advantages when soil microorganisms with different properties are used together [62]. It is also conceivable that the observed favorable benefits are attributable to the expansion of the mycorrhizal niche in the environment [56,63]. 

The temporal patterns of AMF alpha diversity and community composition were not affected by inoculation treatment or cropping patterns. This might be due to AMF’s reproductive phenology being strongly constrained by evolutionary limitations [64]. Additionally, AMF diversity increased significantly in the second stage, which could be due to the soil nutrient depletion caused by plant consumption during this period. Traits that allow for early colonization of host plants, such as the generation of more AMF spores, may have significant tradeoffs with the reserved competitive ability [65]. Alternatively, host plants may preferentially transfer incentives (in the form of increased photosynthate allocation) to more advantageous partners, resulting in changes in AMF community temporal dynamics over time [66]. We found that neither cropping pattern nor AMF inoculation disrupts the temporal dynamics of AMF community composition. The findings indicate that AMF inoculation has no effect on the native AMF community composition and that the manner of regulation of soil AMF community composition may be influenced by complex environmental conditions that promote orderly succession through several growth stages.

In the network analysis, we observed that the nodes of some AMF taxa, e.g., *Diversispora* and *Claroideoglomus* in the inoculation and control treatments, respectively, increased throughout the growth stage. *Glomus* species’ dominance might be attributed to their environmental adaptability, host specificity, functional significance, or ease of reproduction in the soil environment [31,67,68]. 

Intercropping is a practice for improving soil fertility and crop yield. When maize and soybeans are intercropped, maize has a substantial competitive advantage for soil nutrition over soybeans, resulting in changes in soil parameters [69]. Subterranean root–root interactions between intercropped crops might result in a heterogeneous distribution of nitrogen in the soil profile, hence increasing N input into the cropping system through symbiotic N fixation [70]. However, our investigation found that cropping pattern had a substantial effect on the NH_4_^+^–N and NO_3_^−^–N content, which is consistent with earlier research [71]. Changes in aboveground plant diversity can modify soil characteristics, hence affecting microbial diversity in intercropping systems [72,73]. Numerous studies have shown that intercropping may enhance the N, K, and TOC levels of the soil [74,75]. Furthermore, several variables, such as soil type, soil condition, plant species and nutrition, have been observed to influence soil fungus diversity [76]. As a result, intercropping systems have an effect on soil fungal diversity, which may alter in response to variations in plant variety and soil fertility [77].

## 5. Conclusions

To explore the feasibility and sustainability of agricultural development on coal mine dump land, we determined whether AMF inoculation and cropping patterns could affect native AMF communities and the yield of maize and soybean. Our results demonstrated that AMF diversity was significantly influenced by cropping pattern and growth stage, but not by AMF inoculation. Notably, AMF inoculation altered the native AMF community composition in the early growth stage, leading to a more complex AMF network structure in the early and late growth stages. These results indicate that the effect of AMF inoculation on native AMF may only exist in the early stage, and its impact on crop yield may be the consequence of cumulative effects due to the advantages of plant growth and nutrient uptake in the early stage. Future research will focus on the impact of biofertilizers on the nutritional quality of crops and their microbiochemical processes in the soil.

## Figures and Tables

**Figure 1 ijerph-19-17058-f001:**
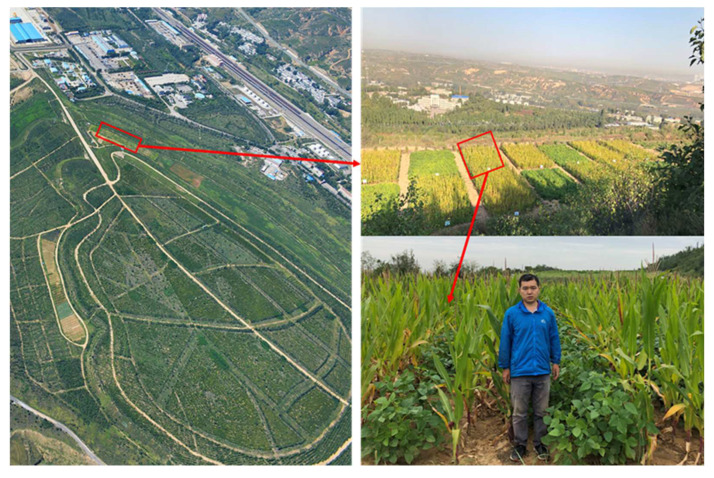
Experimental design.

**Figure 2 ijerph-19-17058-f002:**
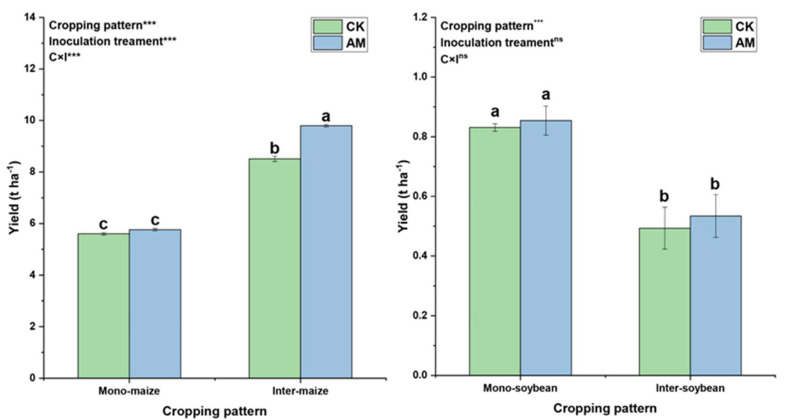
Yield under different cropping patterns and inoculation treatments in the harvesting stage. Mono− and inter−maize were *Zea mays* L. ‘Chenghai No. 618’ in monocropping and intercropping, mono− and inter−soybean were *Glycine max* L. ‘Zhonghuang No. 17’ in monocropping and intercropping. Standard deviation is shown by the deviation bars. Significantly different are bars followed by lowercase letters (*p* < 0.05). ns: *p* > 0.05, ***: *p* < 0.001.

**Figure 3 ijerph-19-17058-f003:**
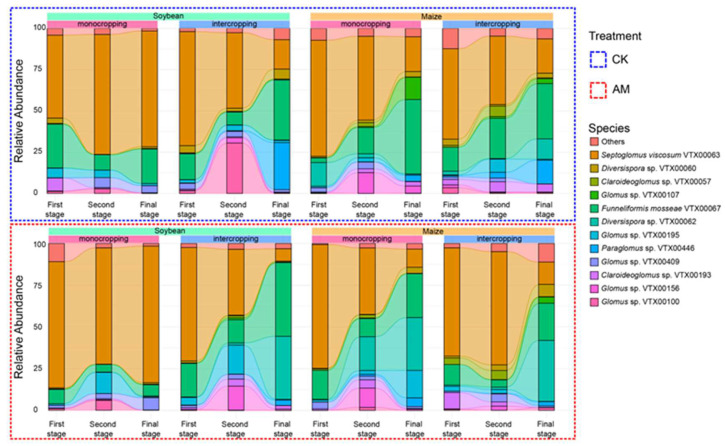
Column chart of the proportion of AMF species.

**Figure 4 ijerph-19-17058-f004:**
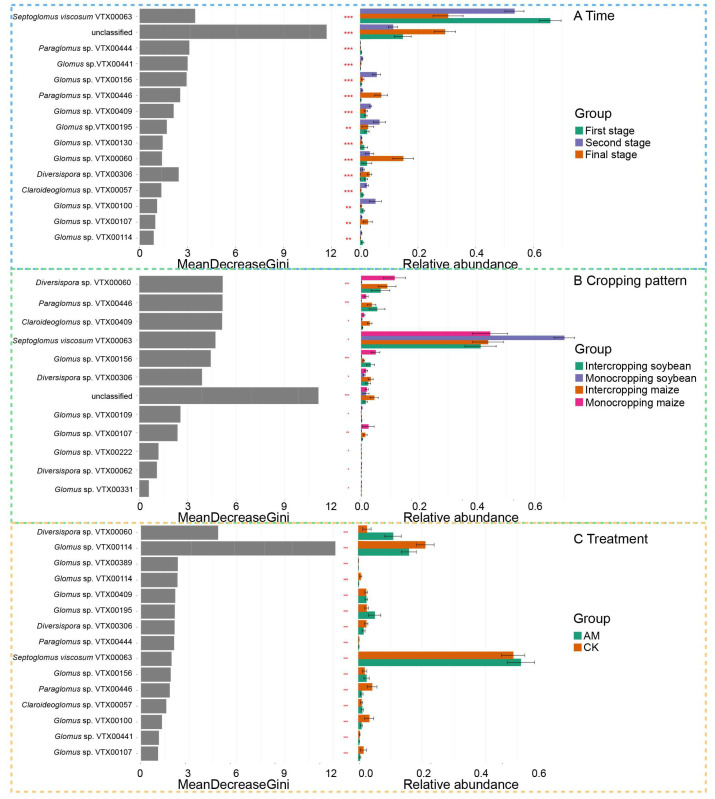
Random forest analysis showing bias distribution of the relative abundance of AMF taxa. (**A**) distribution in different growth stages, (**B**) distribution in different cropping patterns, (**C**) distribution in different inoculation treatments. *: *p* < 0.05, **: *p* < 0.01, ***: *p* < 0.001.

**Figure 5 ijerph-19-17058-f005:**
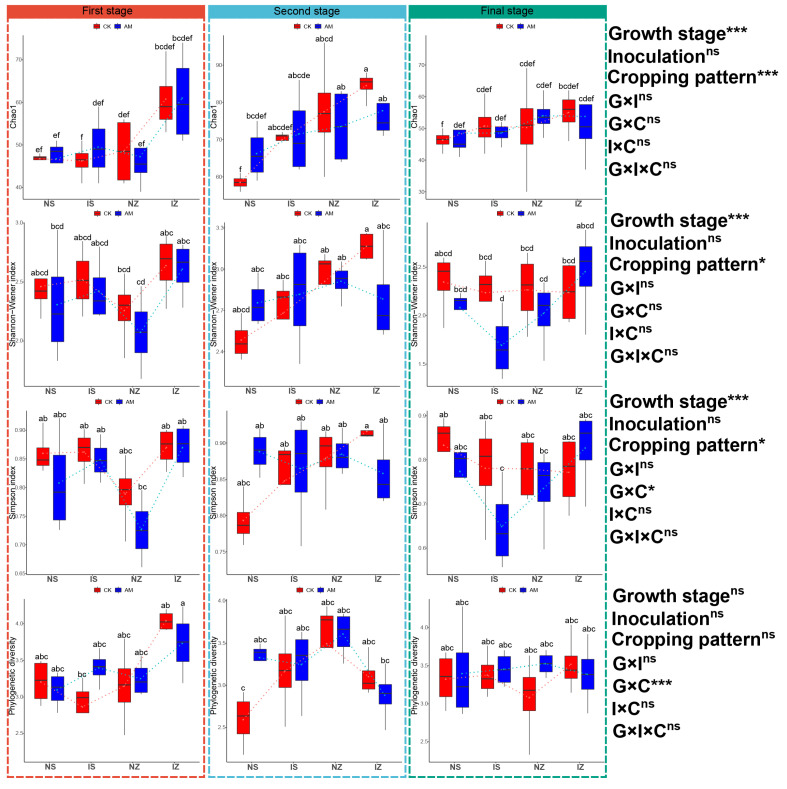
Differences in fungal Chao1, Shannon–Wiener index, Simpson index, and phylogenetic diversity under different treatments. Deviation bars represent standard deviation. Bars followed by lowercase letters are significantly different (*p* < 0.05). ns: *p* > 0.05, *: *p* < 0.05, ***: *p* < 0.001.

**Figure 6 ijerph-19-17058-f006:**
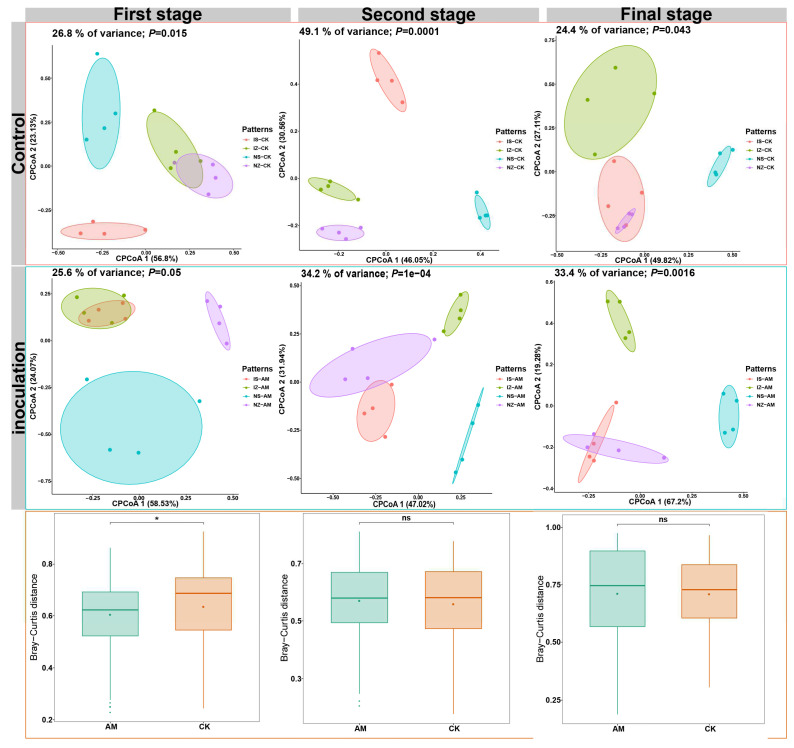
Principal coordinate analysis based on Bray–Curtis dissimilarities of AMF community compositions at different cropping patterns and plant growth stages.

**Figure 7 ijerph-19-17058-f007:**
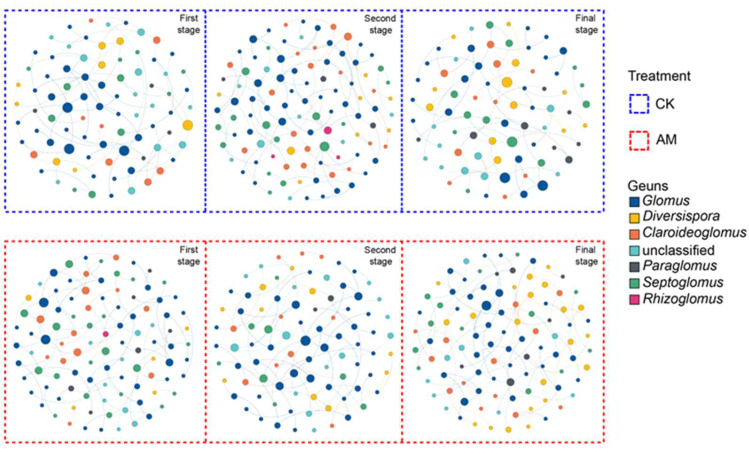
Overview of the networks under different growth stages and inoculation with node sizes being proportional to node degrees.

**Figure 8 ijerph-19-17058-f008:**
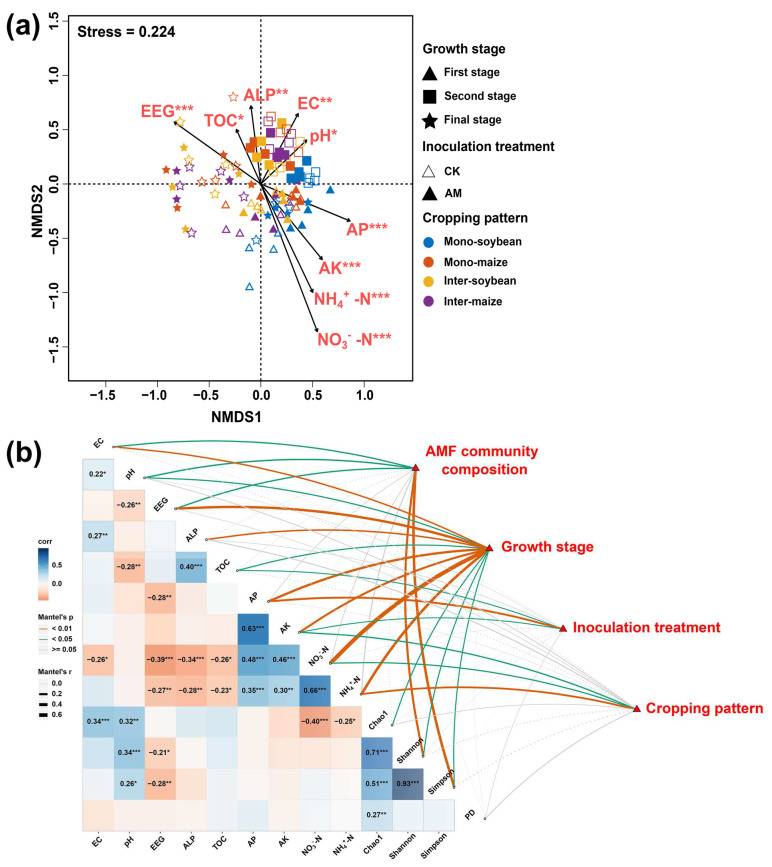
(**a**) Nonmetric multidimensional scaling (NMDS) of fungal community. The arrows represent fitted vectors of edaphic variables with distributions that are significantly correlated with AMF community composition (* *p* < 0.05, ** *p* < 0.01, *** *p* < 0.001). (**b**) Relationships among growth stage, inoculation treatment, cropping pattern, edaphic variables, AMF diversity indices and AMF community composition. Pairwise comparisons of fungal diversity indices and edaphic variables, with a gradient of colors denoting Pearson’s correlation coefficient. (* *p* < 0.05, ** *p* < 0.01, *** *p* < 0.001). Growth stage, AMF community composition, cropping pattern and inoculation treatment were related to fungal diversity indices and edaphic variables by the Mantel test. TOC: soil total organic, AP: available phosphorus, AK: available potassium, EEG: easily extractable glomalin, PD: phylogenetic diversity, Shannon: Shannon–Wiener index.

**Table 1 ijerph-19-17058-t001:** Three-way ANOVA showing the effect of growth stage, inoculation treatment, cropping pattern, and their interactions on electrical conductivity (EC), pH, easily extractable glomalin (EEG), Alkaline phosphatase (ALP), Total organic C (TOC), Available P (AP) available K (AK), NO_3_^−^-N and NH_4_^+^-N.

Source of Variation	Growth Stage (G)	Inoculation Treatment (I)	Cropping Pattern (C)	G × I	G × C	I × C	G × I × C
EC	<0.001	≥0.05	<0.001	<0.001	<0.001	<0.001	<0.001
pH	<0.01	<0.05	≥0.05	<0.01	<0.001	≥0.05	<0.001
EEG	<0.001	≥0.05	<0.001	<0.01	<0.001	≥0.05	<0.05
ALP	<0.001	≥0.05	<0.001	≥0.05	<0.001	<0.01	<0.05
TOC	<0.05	<0.01	<0.05	≥0.05	<0.001	<0.001	<0.001
AP	<0.001	<0.001	<0.001	<0.001	<0.001	<0.001	<0.001
AK	≥0.05	<0.001	<0.001	≥0.05	≥0.05	<0.001	≥0.05
NO_3_^−^-N	<0.001	<0.05	<0.001	<0.01	<0.001	<0.01	<0.001
NH_4_^+^-N	<0.001	≥0.05	<0.001	<0.01	<0.001	<0.001	<0.001

## Data Availability

Not applicable.

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
