# Peer review of "AMF Inoculum Enhances Crop Yields of Zea mays L. ‘Chenghai No. 618’ and Glycine max L. ‘Zhonghuang No. 17’ without Disturbing Native Fugal Communities in Coal Mine Dump"

_ijerph, 2022, doi:10.3390/ijerph192417058_

Round 1

Reviewer 1 Report

The work, delaing with the effect of AMF inoculum on crop yiled is  significant, since all the nature-beased approaches on plant production are of hi-gh importance nowadays.

The Introduction part is thorough and well-organised.

lines 88-93 please provide arguments to support the objectives. Is the aim to enable arable crops to be grown on mine sites, or is the long-term aim of the experiment to clean up (for phytoremediation) already contaminated sites?

What is the intended consumption of the crops produced? Feed and/or human consumption?

Please explain the above in order to better understand the objectives and rewrite the last paraghraph of the introduction accordingly.

Materials and metods are adequately chosen.

lines 102-106 Please include a table of the meteorological data

lines 427-428 consistent with which earlier research?

lines 445-446 Please include the continuation directions and explicit future perspectives of the reserach

Author Response

The work, delaing with the effect of AMF inoculum on crop yiled is significant, since all the nature-beased approaches on plant production are of high importance nowadays.

Response: We have tried our best to revise our manuscript in statistical approach. We hope that the revised manuscript is now acceptable for publication.

The Introduction part is thorough and well-organised.

Response: Thanks for your comments.

lines 88-93 please provide arguments to support the objectives. Is the aim to enable arable crops to be grown on mine sites, or is the long-term aim of the experiment to clean up (for phytoremediation) already contaminated sites? What is the intended consumption of the crops produced? Feed and/or human consumption? Please explain the above in order to better understand the objectives and rewrite the last paraghraph of the introduction accordingly.

Response: Thanks for the suggestions of reviewer, we have added “This study aims to improve crop yields and soil fertility using biofertilizers that are beneficial to the environment. The economic, ecological, and social benefits of mine dumps are achieved through the production of green, organic, and healthy human consumption products.” to the last paraghraph of the introduction, please see line 93-96.

Materials and metods are adequately chosen.

Response: Thanks for your comments.

lines 102-106 Please include a table of the meteorological data

Response: Thanks for the suggestions of reviewer. The lack of continuous observation meteorological data at the study location makes it difficult to provide the data in a table for this investigation. As much as possible, we have supplemented pertinent data, such as the soil moisture data in line 109-110. This component will be the focus of future research. Again, many thanks for your suggestion.

lines 427-428 consistent with which earlier research?

Response: Considering the reviewer’s suggestion, we have added relevant references, please see line 438

lines 445-446 Please include the continuation directions and explicit future perspectives of the research

Response: Thanks for the suggestions of reviewer, we have added “Future research will focus on the impact of biofertilizers on the nutritional quality of crops and their microbiochemical processes in the soil.” to conclusions, please see conclusions line 455-456.

Reviewer 2 Report

This study describes the effect of AMF inoculum on the soil of coal mine dump for crop yields. The changes of AMF in soil were systematic description. However, there are still some questions to be clarified.

1.      Line 14. AMF should be full name at the first time

2.      Line 97. Soil classification should be identified.

3.      Line 111. What kind of AMF was inoculated? Species? The inoculated AMF should be described in the materials and methods.

4.      Line 114. 4.66 m.g-1 spores g−1 superscript.

5.      In section 3.3 What is the change of the inoculated AMF in the AMF treatment soil? VTX00063, VTX00060, VTX00057, VTX00107…. These strains were the inoculated AMF? This needs to be clarified.

6.      Line 280. What is Phylogenetic index?

7.      Line 383. What percentage of AMF inoculation was changed?

8.      What is the soil moisture in this study? In general, the soil moisture content will affect the performance of microbes.

Author Response

This study describes the effect of AMF inoculum on the soil of coal mine dump for crop yields. The changes of AMF in soil were systematic description. However, there are still some questions to be clarified.

Response: We have tried our best to revise our manuscript in statistical approach. We hope that the revised manuscript is now acceptable for publication.

  1. Line 14. AMF should be full name at the first time

Response: Thanks for the suggestions of reviewer, we have revised. Please see line 16.

  1. Line 97. Soil classification should be identified.

Response: Thanks for the suggestions of reviewer, we have added “The soil texture is loess and covers the whole coal field.” to 2.1 study area, please see line 102.

  1. Line 111. What kind of AMF was inoculated? Species? The inoculated AMF should be described in the materials and methods.

Response: Considering the reviewer’s suggestion, we have added “The AMF inoculants were created by inoculating sandy soils with Funneliformis mosseae (a type of AMF) spores and extraradical mycelium generated in pot cultures by maize plants.” to 2.2 Experimental Design, please see line 116-118.

  1. Line 114. 4.66 m.g-1 spores g−1 superscript.

Response: Done.

  1. In section 3.3 What is the change of the inoculated AMF in the AMF treatment soil? VTX00063, VTX00060, VTX00057, VTX00107…. These strains were the inoculated AMF? This needs to be clarified.

Response: Thanks for the suggestions of reviewer. The AMF we used for inoculation was Funneliformis mosseae, which has been supplemented in 2.2 Experimental Design. VTX is the virtual species number of MaarjAM database, and we have marked the inoculated AMF that may match. please see line 254.

  1. Line 280. What is Phylogenetic index?

Response: The reviewer’s suggestion is undoubtedly correct. Phylogenetic diversity is a measure of biodiversity which incorporates phylogenetic difference between species. Therefore, we have replaced "Phylogenetic index" in the full text with "phylogenetic diversity".

  1. Line 383. What percentage of AMF inoculation was changed?

Response: Thanks for the suggestions of reviewer. There seems to be some ambiguity here, so we have replaced "AMF inoculation" with "AMF treatment".

  1. What is the soil moisture in this study? In general, the soil moisture content will affect the performance of microbes.

Response: Considering the reviewer’s suggestion, we have added “The range of soil moisture during the crop growth period is 7.2%-23.8%, with a mean value of 13.8%.” to 2.1 study area, please see line 109-110.

Reviewer 3 Report

Dear Authors,

 I find the manuscript submitted for review interesting, important for human health and the environment, well thought out and carefully prepared. It requires only minor changes.

 1.    Title - I propose to change to: AMF Inoculum Enhances  Yield of Zea mays L. ‘Chenghai No. 123 618’ and Glycine max L. ‘Zhonghuang No. 17’, without Disturbing Native Fugal Communities in Coal Mine Dump

2.    Abstract is unqualified. It contains all the necessary elements in the study summary.

3.    Keywords - I propose to change to: AMF community composition; intercropping; coal mining dump; bioinvasive risk; crop yields.

4.    The introduction is nice and succinct. Well, explains the study design.

5.    The study was carried out without methodological errors.

6.    The presentation of data, including results, is scientifically substantiated and meets the relevant reporting standards.

7.    The discussion section contains the appropriate reference of the obtained results to the research of other authors.

8.    The conclusions are  clear and references enough and well selected.

Text Notes:

Figure 1S can be inserted into the text.

L 50 – AMF change to - Arbuscular mycorrhizal fungi (AMF),

L 54 – remove „Arbuscular mycorrhizal fungi’,

L 65 – „[18,19]” change to [18,19],

L102 - 3350°C???,

L 113, 114, 124, 125 – „g-1, ha-1” - apply superscript,

L 123-124 – I propose change to: Zea mays L. 'Chenghai No. 123 618' and Glycine max L. 'Zhonghuang No. 17' were selected for the study.

L143, 144 - Available phosphorus (AP) and available potassium (AK) – change to: Available phosphorus (P) and available potassium (K),

L 216 - Add Latin names and cultivars of species in the title.

L 231, 232 – „Total organic carbon (TOC), Available phosphorus (AP) available potassium (AK)” change to: Total organic C (TOC), Available P (AP) available K (AK),

L 395-397 – „phosphorus and potassium” change to P, K.

Author Response

Dear Authors,

I find the manuscript submitted for review interesting, important for human health and the environment, well thought out and carefully prepared. It requires only minor changes.

Response: We have tried our best to revise our manuscript in statistical approach. We hope that the revised manuscript is now acceptable for publication.

1.Title - I propose to change to: AMF Inoculum Enhances Yield of Zea mays L. ‘Chenghai No. 123 618’ and Glycine max L. ‘Zhonghuang No. 17’, without Disturbing Native Fugal Communities in Coal Mine Dump

Response: Thanks for the suggestions of reviewer, we have revised. Please see line 1-5.

  1. Abstract is unqualified. It contains all the necessary elements in the study summary.

Response: Thanks for your comments.

3.Keywords - I propose to change to: AMF community composition; intercropping; coal mining dump; bioinvasive risk; crop yields.

Response: As suggested by the reviewers, modifications have been made. Please see line 28-29

4.The introduction is nice and succinct. Well, explains the study design.

Response: Thanks for your comments.

5.The study was carried out without methodological errors.

Response: Thanks for your comments.

6.The presentation of data, including results, is scientifically substantiated and meets the relevant reporting standards.

Response: Thanks for your comments.

7.The discussion section contains the appropriate reference of the obtained results to the research of other authors.

Response: Thanks for your comments.

8.The conclusions are clear and references enough and well selected.

Response: Thanks for your comments.

Text Notes:

Figure 1S can be inserted into the text.

Response: Considering the reviewer’s suggestion, we have inserted “Figure 1S” as “Figure 1”. Please see line 131.

L 50 – AMF change to - Arbuscular mycorrhizal fungi (AMF),

Response: Thanks for the suggestions of reviewer, we have modified. Please see line 52.

L 54 – remove „Arbuscular mycorrhizal fungi’,

Response: Thanks for the suggestions of reviewer, we have modified. Please see line 55.

L 65 – „[18,19]” change to [18,19],

Response: Done.

L102 - 3350°C???,

Response: Thanks for the suggestions of reviewer, this part may cause ambiguity in reading, so we have deleted it.

L 113, 114, 124, 125 – „g-1, ha-1” - apply superscript,

Response: Thanks for the suggestions of reviewer, we have modified.

L 123-124 – I propose change to: Zea mays L. 'Chenghai No. 123 618' and Glycine max L. 'Zhonghuang No. 17' were selected for the study.

Response: Considering the reviewer’s suggestion, we have modified. Please see line 128-129.

L143, 144 - Available phosphorus (AP) and available potassium (AK) – change to: Available phosphorus (P) and available potassium (K),

Response: Considering the reviewer’s suggestion, we have done. lease see line 128-129.

L 216 - Add Latin names and cultivars of species in the title.

Response: Considering the reviewer’s suggestion, we have added Latin names and cultivars of species in the title. lease see line 223-227.

L 231, 232 – „Total organic carbon (TOC), Available phosphorus (AP) available potassium (AK)” change to: Total organic C (TOC), Available P (AP) available K (AK),

Response: Thanks for the suggestions of reviewer, we have modified “Total organic carbon (TOC), Available phosphorus (AP) available potassium (AK)” to “Total organic C (TOC), Available P (AP) available K (AK)”, please see line 238-240.

L 395-397 – „phosphorus and potassium” change to P, K.

Response: Considering the reviewer’s suggestion, we have modified, please see line 404.
